# Emulating Aerosol Optics with Randomly Generated Neural Networks

Andrew Geiss[1], Po-Lun Ma[1], Balwinder Singh[1], and Joseph C. Hardin[1,2]

[1]Pacific Northwest National Laboratory, Richland, WA, USA
[2]Unaffiliated, Santa Clara, CA, USA

**Correspondence:** Andrew Geiss, (andrew.geiss@pnnl.gov)

**Abstract.** Atmospheric aerosols have a substantial impact on climate and remain one of the largest sources of uncertainty in climate prediction. Accurate representation of their direct radiative effects is a crucial component of modern climate models. Direct computation of the radiative properties of aerosol populations is far too computationally expensive to perform in a climate model however, so optical properties are typically approximated using a parameterization. This work develops artificial neural networks (ANNs) capable of replacing the current aerosol optics parameterization used in the Energy Exascale Earth System Model (E3SM). A large training dataset is generated by using Mie code to directly compute the optical properties of a range of atmospheric aerosol populations given a large variety of particle sizes, wavelengths, and refractive indices. Optimal neural architectures for shortwave and longwave bands are identified by evaluating ANNs with randomly generated wirings. Randomly generated deep ANNs are able to outperform conventional multi-layer perceptron style architectures with comparable parameter counts. Finally, the ANN-based parameterization produces significantly more accurate bulk aerosol optical properties than the current parameterization when compared to direct Mie calculations using mean absolute error. The success of this approach makes possible the future inclusion of much more sophisticated representations of aerosol optics in climate models that cannot be captured by extension of the existing parameterization scheme, and demonstrates the potential of random wiring based neural architecture search in future applications in the Earth Sciences.

## 1 Introduction

Atmospheric aerosols have a profound impact on atmospheric radiation, and ultimately the entire Earth system, both through their direct radiative effects (Hansen et al., 2005; Johnson et al., 2018) and interaction with clouds (Twomey, 1977; Albrecht, 1989; Fan et al., 2016). They have long been known as one of largest sources of internal uncertainty in climate modeling, primarily due to cloud interactions, but with a significant contribution from direct effects as well (Bellouin et al., 2020). Difficulties arise in both accurately modeling aerosol populations (Liu et al., 2012) and in determining their subsequent impacts in these areas. While in many cases the underlying physics may be well understood, modeling complex small-scale processes is not computationally feasible within an Earth System Model (ESM), and instead these key physical processes are represented by parameterization schemes.

Recently, there has been a flurry of research that has leveraged new advances in machine learning (ML) to enhance climate and weather modeling (Boukabara et al., 2021). Various strategies have been used, including: emulation of an entire weather or climate model (or at least key fields) with deep learning (Scher, 2018; Weyn et al., 2020), nudging parameterization output (Watt-Meyer et al., 2021; Bretherton et al., 2022), enhancing model output (Wang et al., 2021; Geiss et al., 2022), replacing key model physics such as the radiative transfer scheme (Krasnopolsky et al., 2012; Lagerquist et al., 2021), and replacing the many parameterizations that approximate un-resolvable sub-grid scale processes (Krasnopolsky et al., 2013; Rasp et al., 2018; Brenowitz and Bretherton, 2018). While many of these approaches have some overlap, most are not mutually exclusive strategies for improving climate forecasts: conventional climate models must be used to generate training data for purely data-driven ML models (e.g. Gettelman et al. (2021)) and, in the future, those physics-based ESMs may be significantly enhanced by replacing key parameterization schemes with ML-emulators for instance. Ideally, future climate models will leverage continued research in model development in conjunct with multiple ML-based approaches to generate climate simulations with unprecedented accuracy.

This research focuses on developing an Artificial Neural Network (ANN) emulator to replace the current aerosol optics parameterization developed by Ghan and Zaveri (2007) for internally mixed aerosols represented by the 4-mode version of the Modal Aerosol Module (MAM4) (Liu et al., 2016) in the Energy Exascale Earth System Model's (E3SM) (Golaz et al., 2019) Atmosphere Model (EAM) (Rasch et al., 2019). We perform a thorough neural architecture search using randomly generated ANN wirings to identify ANN structures best suited to accurately representing aerosol optics with the fewest possible parameters (i.e. at the lowest computational cost). Finally, we show that the ML-based parameterization can significantly outperform the current parameterization in terms of accuracy, and even outperforms very high-resolution aerosol optics lookup tables, which would be too large to use in EAM, but can be seen as a high-resolution extension of the current parameterization.

Section 2 of this paper provides background information on the radiative effects of atmospheric aerosols and the aerosol optics parameterization currently used in E3SM. Section 3 discusses how training and testing datasets were generated and how the neural network input and output variables are handled. Section 4 describes the randomly generated ANN approach in detail and the network training procedure and evaluation of the neural architectures. Section 5 evaluates the accuracy of the final ML-based parameterization. Finally, Section 6 provides a short summary of results and some concluding remarks.

## 2  Background

### 2.1  Modeling radiative effects of atmospheric aerosols

Atmospheric aerosols influence Earth's radiative budget both through direct interactions with radiation and modification of clouds (Boucher et al., 2013). Both effects have long been major sources of uncertainty in climate simulations as chronicled by over three decades of assessment reports from the Intergovernmental Panel on Climate Change (see Bellouin et al. (2020) Table 1). Accurate representation of atmospheric aerosols in climate simulations is hindered by many challenges, including complex aerosol-chemical and microphysical processes, aerosol-cloud-precipitation interactions, and aerosol-radiation interactions. Even though the underlying physics have been studied in great detail and accurate physics and theory-based models

exist to represent the relevant processes, these models are far too computationally expensive to use in an ESM. Instead, such processes are represented with simplified physical models and parameterizations that usually make sweeping simplifications in their representation of aerosol processes and trade model accuracy for computational tractability.

One crucial component of an atmospheric model is a radiation scheme. Radiative transfer models are responsible for representing the radiative exchange of energy between space, the Earth's surface, and the many intervening layers of the atmosphere resolved by an ESM. The radiative flux divergence computed by radiation code is used to determine heating rates in the atmosphere which ultimately impact large-scale atmospheric dynamics. E3SM uses the version of the Rapid Radiative Transfer Model (RRTM) (Mlawer et al., 1997; Mlawer and Clough, 1997) developed for use in general circulation models (RRTMG)

(Iacono et al., 2008; Pincus and Stevens, 2013). RRTMG does not take information about aerosol populations as a direct input. Instead, the bulk optical properties of the aerosol populations in each grid-cell are first estimated using a parameterization scheme (Ghan and Zaveri, 2007), and these properties (bulk absorption, extinction, and asymmetry parameter) are passed to the radiative transfer scheme.

   Estimation of the optical properties for aerosol populations in each model grid-cell is, on its own, a computationally daunting

task. Scattering of light by particles is generally separated into three regimes that are defined by the ratio between the radius of the particle ($r$) and the wavelength of light ($\lambda$): Rayleigh ($r << \lambda$), Mie ($r \approx \lambda$), and geometric ($r >> \lambda$). In both the Rayleigh and geometric scattering regimes the optical properties of an aerosol particle vary smoothly as a function of its size. In the Mie regime however, absorption and scattering efficiencies can vary wildly as a function of changing particle diameter. Mathematically, these undulations arise as the solution to Maxwell's equations applied to propagation of electromagnetic

radiation over a spherical particle (Van de Hulst, 1957). A significant portion of atmospheric aerosols have size parameters ($x = 2\pi r/\lambda$) within the Mie regime, particularly in the shortwave radiative bands used by EAM's radiative transfer code. There is no strict definition of the bounds of the Mie regime, but typically one would use Mie code to estimate optical properties for size parameters within about 2 orders of magnitude of unity and geometric or Rayleigh approximations for larger or smaller particles (respectively) depending on the accuracy required for the application (Bohren and Huffman, 1983). Here, we use a

Rayleigh approximation for size parameters less than 0.05 and Mie code for everything larger. Mie scattering solutions can be found in the form of an infinite series, though these series are weakly converging, and sometimes require a large number of terms to accurately determine a particle's optical properties (Hansen and Travis, 1974; Bohren and Huffman, 1983). This is a scenario where the underlying physics are understood and accurate numerical models to represent the physics have been developed (Wiscombe, 1979; Sumlin et al., 2018), but they are far too computationally expensive to use at a large scale, and

parameterizations must be used to represent this physics in an ESM (Ghan and Zaveri, 2007; Pincus and Stevens, 2013). This parameterization must represent a high-dimensional manifold in a space defined by the parameters of the aerosol size distribution, the imaginary and real components of aerosol refractive indices (which depend on the aerosol species), and various wavelengths of light. The portion of this manifold that falls in the Mie regime is characterized by large fluctuations, particularly with respect to wavelength and particle size, and any function used to parameterize it will likely require a large number of

parameters to adequately capture this variability. In this work, we focus on developing a parameterization of bulk aerosol radiative properties that is fast enough to use in an ESM and substantially more accurate than previous methods.

## 2.2 E3SM and the Modal Aerosol Module (MAM4)

This study focuses on updating the aerosol optics representation for E3SM, an ESM developed by the U.S. Department of Energy (Golaz et al., 2019). EAMv1 (Rasch et al., 2019) uses the 4-mode version of the Modal Aerosol Module (MAM4) (Liu et al., 2012, 2016) with improvements to represent aerosol processes (Wang et al., 2020), RRTMG for atmospheric radiative transfer (Iacono et al., 2008; Pincus and Stevens, 2013), and the Ghan and Zaveri (2007) parameterization for aerosol optics. This parameterization is also used in other ESMs, including the Community Earth System Model v2.2 (Danabasoglu et al., 2020; NCAR, 2020), so the new parameterization developed in this study can be easily used in other ESMs.

MAM is a simplified model of aerosol populations that was developed to allow representation of key aerosol physics in climate simulations without being computationally prohibitive. Because of the complexity of the general dynamic equation for aerosols (Friedlander, 2000), several methods for representing aerosols in simulations of the atmosphere exist that have varying degrees of accuracy and computational complexity. These include bulk models (Lamarque et al., 2012), modal models (Liu et al., 2012), the sectional method (Gelbard et al., 1980), the quadrature method of moments (McGraw, 1997), and discrete models (Gelbard and Seinfeld, 1979). The key differences between these models are primarily their treatment of aerosol size distributions and mixing. Section 1 of Liu et al. (2012) and Table 1 of Zhang et al. (2020) provide overviews of different approaches to modeling aerosol populations.

The MAM approach breaks aerosols down into several modes based on species and approximate size. MAM4 includes Aitken, Accumulation, Coarse, and Primary Carbon modes. Each mode contains multiple aerosol species within a certain particle size range and MAM assumes internal mixing within modes and external mixing between modes (aerosol properties are averaged within each mode). The modal model assumes that the size distributions of each mode are log-normal and prescribes the log-standard deviations based on past observational studies. Major uncertainty in the modal approach stems from the limited representation of internal vs external mixing of aerosol species and the assumption of log-normal size distributions. It is reasonably accurate and very computationally efficient compared to other schemes however, and this makes it a good choice for long-duration ESM simulations.

## 2.3 The Ghan and Zaveri (2007) aerosol optics parameterization

EAMv1 uses a parameterization to estimate the bulk optical properties of simulated aerosols. The parameterization is described in detail in Ghan and Zaveri (2007) with further relevant information found in Ghan et al. (2001) and Neale et al. (2012), but we will provide a brief overview of the method here because it will be useful for understanding subsequent sections of this paper. A diagram of the aerosol optics parameterization training/preparation and how it integrates with EAMv1 is provided in Figure 1 and may be a helpful reference while reading this section.

The existing optics parameterization estimates optical properties based on five input parameters: aerosol mode (corresponding to MAM modes), wavelength band ($\lambda$), real refractive index ($n$), imaginary refractive index ($\kappa$), and mean surface mode radius ($r_s$). Optical properties are pre-computed over a range of values in each of these five dimensions, and when called by the

model, the parameterization estimates optical properties from these pre-computed values using a combination of Chebyshev and linear interpolation.

The pre-computed optical properties are generated as follows: for each wavelength band and aerosol mode, refractive index bounds are computed by taking the minimum and maximum refractive indices across all aerosols in that mode and water. The real refractive index range is spanned by 7 linearly spaced values and the imaginary refractive index range is spanned by 10 logarithmically spaced values. Then a range of 200 plausible aerosol radii is generated between $0.001\mu m$ and $100\mu m$. The wavelength, refractive index, and radii data are fed to a Mie code (Wiscombe, 1979) to compute the optical properties for individual particles. Ultimately the parameterization uses bulk optical properties integrated over a size distribution however, so a range of 30 log-normal size distributions is assumed and the individual particle optical properties are integrated over these size distributions. The size distributions are generated for $r_s$ values between $0.01\mu m$ and $25\mu m$ and spaced according to Chebyshev nodes. The optical properties are then fit with a 5th-order Chebyshev polynomial along the $r_s$ dimension and the 5 Chebyshev coefficients are saved rather than directly saving 30 optical property values (Vetterling et al., 1988). Ultimately a 3-dimensional matrix (real refractive index, imaginary refractive index, and surface mode radius) of Chebyshev coefficients is stored for each wavelength and aerosol mode combination, and four of these must be produced representing the four required output variables: bulk shortwave absorption efficiency, bulk shortwave extinction efficiency, bulk shortwave asymmetry parameter, and bulk longwave absorption efficiency. Because of its high dimensionality, the size of the data stored by the parameterization grows rapidly as the resolution with which it resolves the input parameters is increased. This is a major motivation for replacing the current parameterization with a neural network, because increasing accuracy by increasing resolution of the input parameter space rapidly becomes intractable in the existing parameterization.

When the optics parameterization is called by EAM, it is passed values of $r_s$, $n$, and $\kappa$ for each aerosol-mode/wavelength-band combination. The parameterization applies bi-linear interpolation along the refractive index dimensions of the table to estimate Chebyshev coefficients at an intermediate refractive index. Then, the 5th order Chebyshev polynomial generated with these coefficients is used to estimate the optical properties as a function of $r_s$. This approach is very similar to using a lookup table, in that the optical properties have been pre-computed, with the major difference being that a combination of bi-linear and Chebyshev interpolation is used to resolve three of the dimensions as continuous functions of the input variables.

Errors are introduced at nearly every step in this process, including averaging of within-mode refractive properties, a limited number of wavelength bands treated by the model, assumed aerosol size distributions, interpolation of refractive indices and particle size distributions, and others. This approximation of well understood but un-resolvable physics is a frustrating but unavoidable facet of climate modeling. Here, we set out to replace the Chebyshev interpolation approach with a neural network emulator, which addresses the errors incurred by coarsely resolving $n$, $\kappa$, $r_s$, and particle radius information (evaluated in more detail in Table 1 in Section 5).

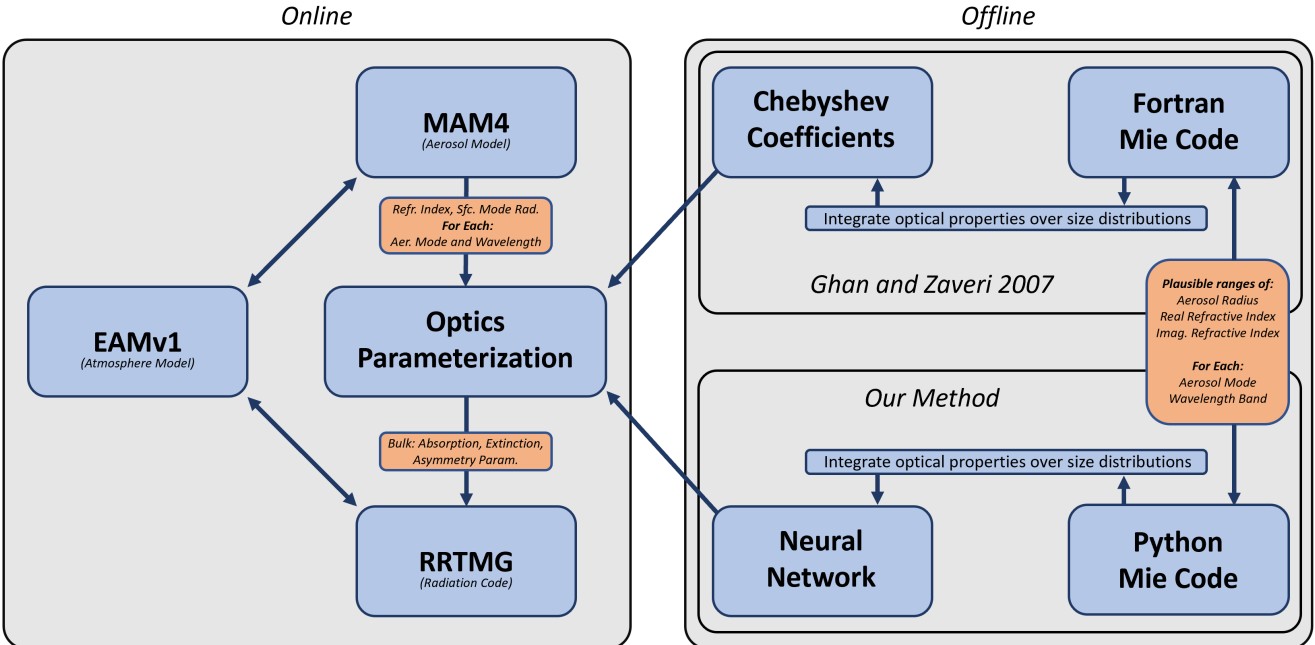

**Figure 1.** A diagram of the aerosol optics parameterization and how it integrates with EAM. The "online" section shows how the parameterization is used during a simulation and the "offline" portion shows the process of pre-computing optical properties and preparing the parameterization.

## 3 Data

### 3.1 Mie code

Training a neural network to emulate Mie scattering first required generation of large training, validation, and testing datasets using established Mie solvers. We chose to refactor the Fortran code used to generate the existing parameterization's pre-computed optical properties into Python. The FORTRAN77 "miev0" Mie scattering code (Wiscombe, 1979, 1980), which was originally used to perform Mie calculations to generate the current EAM parameterization, was replaced by PyMieScatt (Sumlin et al., 2018), a Python-based Mie code. The machine learning libraries used in this study are also written in Python and this refactoring allowed for an end-to-end Python-based pipeline for creating the neural network emulator and will allow for easier and more flexible editing if new training data needs to be generated in the future. Furthermore, PyMieScatt has support for additional scattering models, such as core-shell optics, which we intend to integrate into the neural network emulator in the future. We have made all of the code written for this study available on the project's Github repository (see the "Code and Data Availability" statement).

To ensure that using PyMieScatt did not introduce any additional errors or discrepancy with the original parameterization we performed a comparison to miev0. The optical properties of every refractive index, particle size, and wavelength combination

used by the original parameterization were output and compared to the same optical properties computed using PyMieScatt. The maximum, 99.9th-percentile, and 99th-percentile absolute errors are shown in Table A1. Even the most extreme discrepancies between the two schemes are negligible compared to other sources of error in the parameterization.

## 3.2 Training and validation data

For ANN training, we generated a large table of bulk aerosol optical properties similar to what is described in Section 2.3, but with significantly higher resolution in terms of its input variables. We used the same bounds for possible real and imaginary refractive index values, particle radii, and surface mode radius as in Ghan and Zaveri (2007), and similarly used logarithmic vs linear spacing depending on the variable (the same wavelength bands and aerosol modes were used). The resolution of each of these variables was increased to 2049 particle radii, 257 mode radii, 129 imaginary refractive indices and 129 real refractive indices. This is in comparison to 200, 30, 10 and 7 values (respectively) in the original parameterization. The resulting high-resolution table has about 20,000 times the number of entries, takes on the order of 1-day to compute using parallelized calls to PyMieScatt on a modern CPU, and occupies several GB of RAM, making it inappropriate for direct use in an ESM.

When training a neural network, it is best practice to evaluate the ANN on a hold-out set of validation data after it is trained as a check for over-fitting to the training data. The validation data used here were drawn randomly from the high resolution table using half of the data points for training and half for validation. In this application, the boundaries of the optical property tables were chosen by Ghan and Zaveri (2007) to encompass all possible input values the parameterization could receive from the ESM, so we are not concerned about poor performance when extrapolating outside of the optics table. There is potential for over-fitting to cause unexpected behaviour in the regions between points in the training set however, and this choice of validation set allows for detection of this type of over-fit if it occurs.

## 3.3 Testing data

In addition to a validation set, when hyperparameter tuning is used or multiple models are tested, an additional set of "test" data should be held out to ensure that the validation set has not been over-fit by the hyperparameter or model selection (Murphy, 2012). The test set used in this study was generated separately from the training data, and is approximately the same size as the combined training and validation sets. The training set was constructed by generating an additional table of optical properties where each of the input parameters bisects the grid of values used to generate the training and validation data. This ensures that it completely covers the range of possible inputs and does not contain values near any of the training and validation data points. This test set was used to ensure that the randomly wired ANN approach did not lead to an overfit of the validation set.

## 3.4 Benchmark datasets

In addition to the high-resolution optics data used for training and validation, three other tables of optical properties were generated at intermediate resolutions of: $1025 \times 129 \times 65 \times 65$, $513 \times 65 \times 33 \times 33$, and $257 \times 65 \times 17 \times 9$. Where the table dimensions have been listed in the order: (particle radii) $\times$ (mode radii) $\times$ (imaginary refractive index) $\times$ (real refractive

index). We have chosen to scale dimensions to a power of two plus one so that grid points in a table will be bisected by grid points in the next highest resolution table. These datasets have total parameter counts of approximately $10^8$, $10^7$ and $10^6$ respectively once the multiple wavelengths, aerosol modes, and output parameters are accounted for. Note that the number of particle radii used to resolve the particle size distributions does not add to the size of the optics table and is only used when the dataset is generated, but is important to the table's accuracy. The total parameter count, in the shortwave table for example, is computed as: (number of mode radii) × (number of imaginary refractive indices) × (number of real refractive indices) × (14 shortwave bands) × (4 aerosol modes) × (3 optical properties). These additional optics tables were evaluated by linearly interpolating their entries to query points in the test set described above, and the resulting errors are shown in Table 1 in Section 5. They provide an indication of how the resolution of the training data might impact the accuracy of the trained neural network parameterization.

## 3.5 Neural network inputs and outputs

To compute the bulk optical properties of a population of homogeneous spheres with log-normally distributed radii, five values must be known: the real and imaginary components of the refractive index, the geometric mean radius and log-standard deviation that define the size distribution, and the wavelength of light. For the parameterization problem solved here we assist the neural network by encoding this information in a format more conducive to training neural networks.

Neural networks tend to perform better when input and output data have certain well-behaved distributions and formats. Several pre- and post-processing steps were used alongside the ANN to help ensure optimal performance. Each ANN has 9-inputs (in order): $\lambda$, $n$, $\kappa$, $r_s/\lambda$, $r_s$, and a "one-hot" encoding of the four aerosol modes (four values). The one-hot encoding is a common strategy for categorical inputs and usually leads to better performance than a single scalar input that encodes the category (Murphy (2012) p. 35). The existing parameterization prescribes a log standard deviation for each aerosol mode, so the log standard deviation was not included as a separate continuous input. We chose to include $r_s/\lambda$ as a constructed input despite the fact that both of these variables are used as individual inputs because the size parameter is a key input for Mie scattering calculations, and we found this to improve model skill in early experiments. All of the inputs other than the one-hot encoding are scalar and are each standardized by first taking the log (except for real refractive indices where a log is not used), then subtracting the mean and dividing by the standard deviation (each rounded to a precision of 0.1). The means and standard deviations used are shown in Table A2 and are based on data from the training set. This yields dimensionless, zero-centered inputs with a standard deviation of one and without extreme skew or leptokurtosis.

The Ghan and Zaveri (2007) parameterization estimates specific extinction, absorption, and scattering efficiencies, which are bulk optical properties of the aerosol distribution per total wet aerosol mass, but these values can span several orders of magnitude and thus are not well suited for prediction with a neural network. Instead, we have the neural network estimate a key intermediate value used in the Ghan and Zaveri (2007) parameterization that encapsulates the computationally expensive components of estimating bulk aerosol optical properties:

$$\overline{Q} = \frac{1}{\log \sigma \sqrt{2\pi}} \int\limits_0^\infty Q(r, \lambda, m) e^{\left(-0.5 \left(\frac{\log (r/r_s)}{\log \sigma}\right)^2\right)} \frac{1}{r} dr \tag{1}$$

where $\sigma$ is the log standard deviation of the particle size distribution, $r$ is wet particle radius, $\lambda$ is wavelength, $m$ is the complex refractive index, $Q$ is either the extinction or absorption efficiency (see Ghan and Zaveri (2007) Eq. 20), and the over-line indicates a bulk optical property. In MAM, the values of sigma are prescribed for each mode: 1.6 for modes 2 and 4 and 1.8 for modes 1 and 3.

While the values of (1) are constrained to a reasonable range, linear scaling of the outputs of the ANN is still used to ensure that they are bounded by 0 and 1. This allows use of a sigmoid output function to constrain the ANN's outputs. The bulk absorption efficiency is linearly scaled by a factor of 2.2 while the bulk extinction efficiency is scaled by 4.6. These values were determined empirically from the training set and when the parameterization is used in an ESM this scaling will need to be applied. The bulk asymmetry parameter ($\overline{g}$) is naturally bounded by 0-1 for the range of inputs in this study and is not scaled (Bohren and Huffman, 1983). The longwave and shortwave bands have significantly different ranges for some of their inputs, and the existing parameterization only computes bulk absorption in the longwave, so two neural networks were trained, one with three outputs to process the shortwave bands and one with a single output to process the longwave bands.

## 4 Randomly wired neural networks

### 4.1 Neural architecture search

Neural networks are powerful data fitting tools, and simple ANN designs can easily generalize to a wide variety of problems. Even so, specialized ANN architectures that have been optimized for a task will usually perform best. Task-specific ANN design is difficult however, because the space of reasonable ANN designs is usually far too large to explore exhaustively, and it is not usually obvious which will work best. Typically, researchers will rely on heuristics, past experience, or simply convenience and popularity to choose an appropriate ANN architecture.

Various algorithmic approaches to Neural Architecture Search (NAS) (Elsken et al., 2019) and Hyper-Parameter Optimization (HPO) (Hutter et al., 2019) have become popular for addressing this problem. These algorithms usually involve training many different neural networks with a range of parameter and design choices and selecting the best performing models. Search methods range from simple random or grid search to sophisticated algorithms such as evolutionary optimization (Angeline et al., 1994), Bayesian optimization (Bergstra et al., 2013), or reinforcement learning (Baker et al., 2017). Much of the recent (past 10 years) research in neural architecture search has focused on developing new convolutional neural network architectures for image processing (e.g. Zoph et al. (2018)). Elsken et al. (2019) and Yao (1999) provide reviews of this topic.

Most NAS strategies that test a variety of network wiring patterns are limited to exploring certain families of pre-defined network styles, or break up the search space by randomizing individual network "cells" that are then wired together in sequence. Xie et al. (2019) however, demonstrated a NAS strategy in which new convolutional neural network architectures

were discovered through random wiring of network layers. Motivated by early observations during our work that inclusion of skip connections and more complex wirings contributed to performance for the aerosol optics problem, we chose to employ a similar approach here. While Xie et al. (2019) focus on convolutional neural networks, here we use ANNs constructed of fully connected layers. In general, skip connections and complex wirings are much more common in deep convolutional neural network architectures than ones constructed from fully connected layers, but there is some past evidence that including skip connections in deep fully connected networks can improve performance on certain non-linear problems (Lang and Witbrock, 1988), and that seems to be the case for the problem of emulating Mie scattering. Here, we designed an ANN generator that automatically produces ANNs with a random number of layers, random layer sizes, and random connections between layers. Ultimately the randomly generated wirings allow for the discovery of networks that substantially outperform simple multi-layer perceptrons.

## 4.2 Random network motivation

The physical parameterization problem discussed in this paper is particularly well suited for an ANN. The bulk aerosol optical properties used by the parameterization can be thought of as smooth, bounded, manifolds in a high dimensional space, and representing this type of dataset is an area where neural networks often excel. It is also a particularly data-rich problem because the only limits to the size of our training dataset are the computational and storage resources we would like to devote to generating training data (and ultimately an upper bound on training set resolution where neighboring data points become highly autocorrelated). In early experiments, we found that while simple feed-forward multi-layer perceptron style architectures with 1-2 hidden layers can provide much higher performance than the current EAMv1 parameterization discussed in Section 2.3, more complex architectures that included many smaller layers with skip connections could achieve even higher accuracy without an increase in the number of model parameters. Ultimately, when used in a climate model, the ANN-based parameterization will be evaluated many times (every time the radiative transfer code is called for each model grid cell). This means that reducing the network size as much as possible without significantly reducing accuracy is a worthwhile endeavor, and can save both computation-time and memory when the climate model is run. Additionally, because of the relatively small size (500-100,000 parameters) of the ANNs used here, they are cost effective to train. Together, these factors mean that this ML problem is ideal for NAS.

## 4.3 The random ANN generator

Our ANN generator randomizes network layer size, layer count, merge operators, and wiring. First, it randomly chooses a number of layers between 2 and 12, then randomly chooses the number of neurons per layer by chosing an integer between 7 and 45 and scaling it by a factor of $0.5N_{\text{layers}}$ (the scaling prevents generating very deep and wide ANNs with extremely high parameter counts). To facilitate merging inbound tensors to a layer with element-wise addition, all layers in the network use the same number of neurons. Each hidden layer used in the network is a fully connected layer and applies a $tanh$ activation to its outputs.

Once layer counts and size are selected, the ANN generator creates a random wiring between the layers by generating an adjacency matrix that represents layer connections. Several constraints must be enforced on the adjacency matrix to ensure that it represents a usable ANN architecture. Firstly, we require that the ANN is feed-forward. If each row in the adjacency matrix represents a layer in the order in which they will be evaluated in the ANN, this can be accomplished by enforcing the adjacency matrix is lower-triangular. For an ANN with N hidden layers this means there are $\frac{1}{2}(N^2 + N)$ valid layer connections. The number of active connections for an ANN is randomly chosen from a uniform distribution between 0 and $\frac{1}{2}(N^2 + N)$ and then this many entries in the lower triangular portion of the adjacency matrix are randomly turned on. Additionally, each layer must have at least one inbound and one outbound tensor. Because the number of layers in the ANN is determined before the adjacency matrix is constructed, this must be enforced by iterating through each row and column of the adjacency matrix and randomly turning on one valid inbound and/or outbound connection if the corresponding layer has none.

Lastly, the number of inputs to each ANN are static (9-inputs) but we would like the outputs from each network layer to be a fixed size and any layer can be directly connected to the input layer. As a workaround, each ANN includes an additional fully connected layer with a number of neurons equal to the difference between the 9-inputs and the randomly selected network layer size. The outputs from this layer are appended to the actual inputs as a learnable padding.

Initial experiments on a subset of the training data were run using a single shortwave band (because of reduced training time on the smaller dataset) with additional randomizations including: variable layer sizes (ANNs that used different layer sizes internally exclusively used concatenation to merge tensors); randomly selected activation functions from: linear, tanh, rectified linear unit ("ReLU") (Glorot et al., 2011), exponential linear unit (Clevert et al., 2015), Leaky ReLU, and Parametric ReLU (He et al., 2015); and batch normalization (Ioffe and Szegedy, 2015), dropout (Srivastava et al., 2014), or no regularizer. These experiments showed that the $tanh$ function provided slightly better performance than other activations and that including batch normalization or dropout substantially reduced performance. We hypothesize that the reduced performance with dropout is related to the fact that we are testing relatively small networks. Because dropout layers generally force the ANNs to learn redundant representations of the data and the small ANNs used here only have limited capacity to represent the complex training data, requiring them to learn redundant representations of the data only reduces their skill. Additionally, the complexity of the training data and small size of the networks means that we are not particularly concerned about over-fits and do not expect to gain much from using regularization techniques. These additional types of randomizations were not included in final experiments.

### 4.4 Training and model selection

Each model was trained using the Adam optimizer with an initial learning rate of 0.001, $\beta_1 = 0.9$, and $\beta_2 = 0.999$ to optimize mean squared error. We used a batch size of 64 samples and trained for 10 epochs. The learning rate was reduced manually by a factor of 10 on the 4th, 7th, and 10th epochs. 500 randomly wired ANNs were trained and each was evaluated on the validation set. Figure 2 shows scatter plots of each random ANN's validation performance in terms of mean absolute error (MAE) on the standardized ANN outputs plotted against the number of trainable parameters in the network. Both panels in Figure 2 shows a similar pattern in terms of ANN performance versus size: skill improves rapidly with increasing size until it

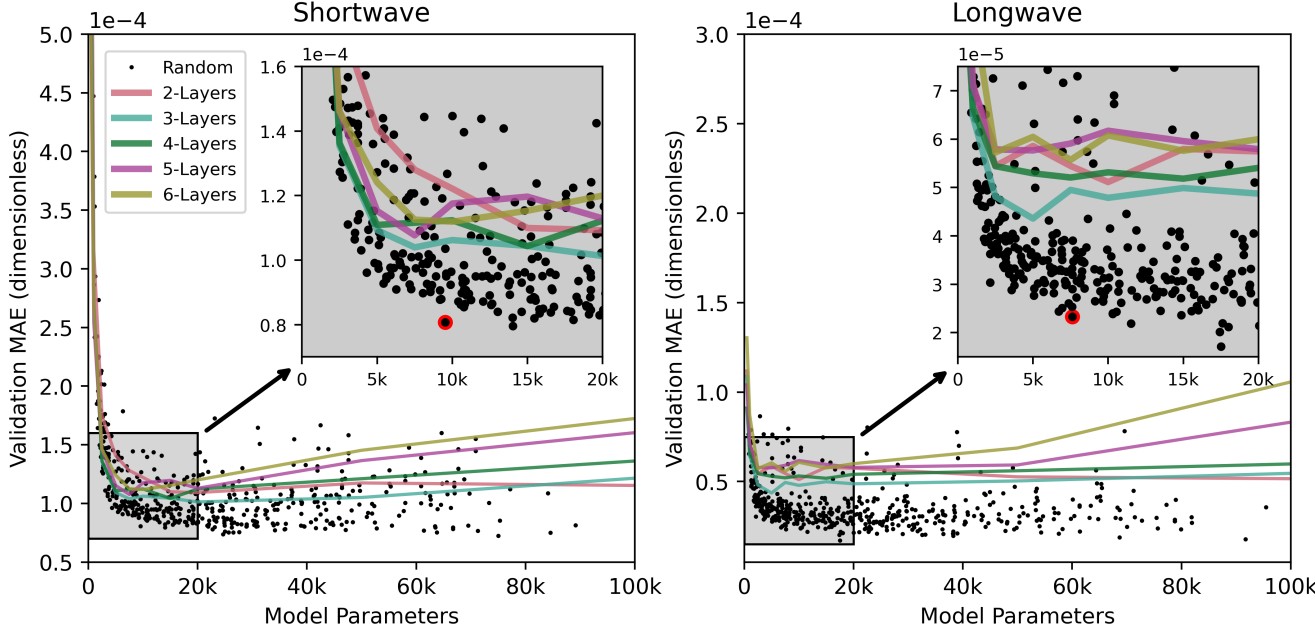

**Figure 2.** Validation set performance of randomly wired neural networks plotted against the network size. Panels show results for different wavelength regimes. The mean absolute error is computed on normalized optical properties (directly on the outputs from the neural networks) and are dimensionless. In each case, there is a clear elbow, beyond which increasing the network size does not substantially improve performance. In both panels, the inset shows a magnified region around this elbow. Solid lines indicate the performance of traditional feed forward multi-layer perceptron ANNs with 2-6 hidden layers. The red dot indicates the network that was ultimately chosen for use.

plateaus somewhere between 1,000-20,000 trainable parameters. Additional size increases yield only very small performance gains. The approximate location of the elbow in each of these performance vs size plots is expanded in an inset panel in each

330 figure panel. Based on these inset plots we subjectively chose an ANN for both the longwave and shortwave regimes that appears to provide a good balance between network size and skill. The selected ANNs are denoted in Figure 2 with red circles, and diagrams of the wirings for the selected networks are shown in Figure 3. Note that later, in Section 5, errors will be reported after re-scaling the standardized network outputs for comparison to the Ghan and Zaveri (2007) scheme, but here we report the selected ANNs' MAEs on the test set computed directly on the ANN output as in Figure 2: SW: $8.96 \times 10^{-5}$, LW: $2.32 \times 10^{-5}$.

The comparable performance on the test set to the validation set indicates that the chosen ANNs did not overfit the training and validation data. These selected ANNs were ultimately retained for use as parameterizations and are evaluated in more detail on the test set in Section 5.

We also trained several benchmark ANNs for comparison to the random ANNs. Each of the benchmark networks is composed of 2 to 6 hidden layers wired in sequence with $tanh$ activation functions and represent the performance of conventional

ANN architectures. 1-layer ANNs performed almost an order of magnitude worse than the others and were not included. Benchmark ANNs with a total of 10 different sizes in terms of total trainable parameters were used. 5-copies of each unique

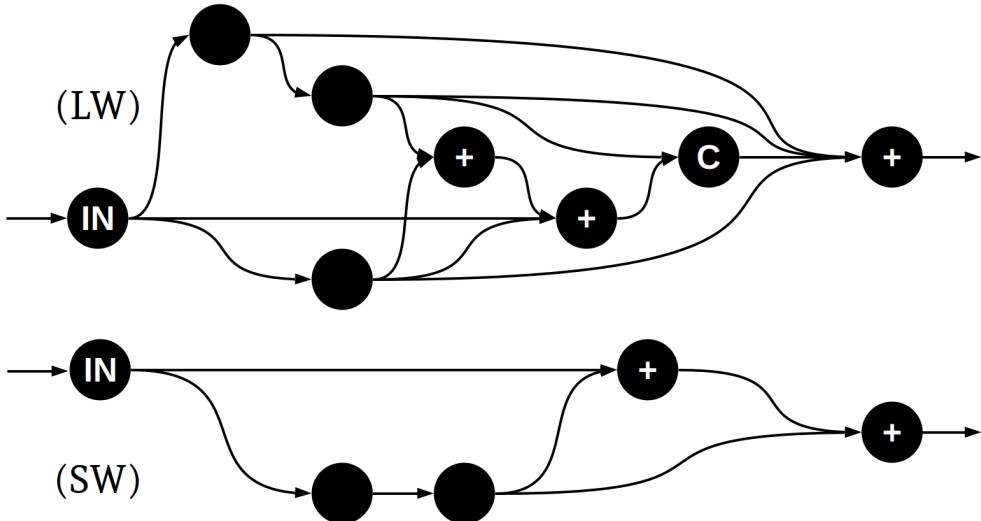

**Figure 3.** Wiring patterns of the two (longwave and shortwave) randomly generated neural networks that were selected for use in the optics emulator. Nodes represent "dense" (fully connected) layers. "C" and "+" indicate whether inbound tensors are combined by concatenation or addition. All hidden layers have the same number of neurons within each network: SW: 54, LW: 32 (the nine inputs are padded to reach the appropriate size and the output layer has either 3 (SW) or 1 (LW) neurons).

benchmark ANN layer-count and parameter-count combination were trained and only the best performing models were retained to ensure that poor performance at a particular ANN size was not simply due to an unlucky random initialization or training sample selection. This means that a total of 250 benchmark ANNs were trained for both the longwave and shortwave regimes. The performance of these benchmark ANNs is also indicated in Figure 2 by solid lines.

### 4.5 Discussion of ANN architecture

The performance of the benchmark and random ANNs provides some insight into ANN design. Firstly, we note that 1-layer ANNs were also tested, but typically performed nearly an order of magnitude worse than other ANNs and are not shown in Figure 2. This suggests that using almost any multi-layer architecture, regardless of construction, can yield substantial performance gains. Secondly, the 2-6 layer sequential models are outperformed by the *majority* of randomly wired ANNs that have similar parameter counts. Also, the multi-layer sequential models with more than 3-layers begin to perform worse than their shallower counterparts. It appears that the inclusion of skip connections has likely allowed the random networks to train successfully despite their depth (high layer count). In the context of this problem, the neural networks are attempting to fit a high-dimensional manifold that varies significantly with respect to several of the input parameters. Deeper networks are likely required to efficiently represent the non-linearities in the problem, but deep neural networks can struggle to train effectively due to vanishing gradients (Goodfellow et al., 2016). The ANNs that were ultimately chosen here tend to have more, but smaller, layers than the best serially connected ANNs, and include multiple skip connections.

The universal approximation theorem implies that this problem is solvable with a wide, single-layer perceptron network (Hornik et al., 1989). In practice however, multi-layer networks are almost always more efficient, and that is the case here. Furthermore, any of the randomly wired networks used here could theoretically be represented by a serially connected multi-layer network: one can imagine a serially connected network learning to apply the identity function to some of its inputs, thereby learning to generate skip connections on its own. Again, while it is technically possible, this is not the case in practice, and even learning the identity function is not necessarily a trivial task for neural networks. While the importance of skip connections has been thoroughly explored in the context of building very deep convolutional neural networks (He et al., 2016) it has only rarely been applied to ANNs with fully connected layers, though some early examples of this approach do exist (Lang and Witbrock, 1988). These results are informative for our application and similar use cases, where the ANN's memory and compute requirements at inference time are of particular importance and by evaluating many ANN architectures we have identified ANNs with significantly higher accuracy than conventional architectures with no increase in inference cost. Taken together, our results indicate that significant performance gains may be achieved in other applications of ANNs in the Earth sciences and Earth system modeling through in-depth exploration of task-optimized network architectures.

## 5 Evaluation

The ANNs were ultimately evaluated on the randomly generated hold-out test set described in Section 3.3. In addition to evaluating the accuracy of their outputs we evaluate them on two additional optical properties derived from the ANN output: shortwave bulk scattering efficiency and single scattering albedo (SSA) which are computed as: $\overline{Q}_{Sca.} = \overline{Q}_{Ext.} - \overline{Q}_{Abs.}$ and $SSA = 1 - \overline{Q}_{Abs.}/\overline{Q}_{Ext.}$ (Bohren and Huffman, 1983). SSA's with $\overline{Q}_{Ext.} < 0.01$ were not included in the analysis because very small errors get amplified by the $\overline{Q}_{Ext.}^{-1}$ in scenarios where scattering is negligible. The existing aerosol optics parameterization was also evaluated along with linear interpolation applied to several high-resolution tables of aerosol optical properties that were generated at a range of resolutions (described in Section 3.4). This includes the very high resolution table used for training and validation. Test set MAE for each of the output parameters and wavelength regimes are listed in Table 1. The ANN shows a substantial performance improvement over the existing parameterization, with MAEs about three orders of magnitude smaller. This is particularly notable for the shortwave extinction efficiencies where the existing parameterization has an MAE of 0.2 but the ANN has an MAE of $3.6 \times 10^{-4}$. Extinction efficiencies range from about 0 to 3.5 so an MAE of 0.2 is substantial. The performance of the additional interpolated optics tables behaves about as expected, with the MAE decreasing in proportion to table size. It can also be seen that to achieve performance comparable to the ANN a lookup table with approximately $10^9$ parameters is required. This is far too large to be used in an ESM. Lastly, Table 1 indicates the test-set performance of the best performing conventional (serially connected) ANN on the test set, and again we see that it cannot match the performance of the randomly wired ANN, which consistently outperforms it by around 10% to 30% for the shortwave and 65% in the longwave.

The very low MAE shown in Table 1 is encouraging, but ideally a parameterization should perform well over the full range of possible inputs and a low MAE could potentially still be achieved in the presence of outlier cases with high error that could cause problems when it is used in a climate simulation. Figure 4 shows logarithmically-scaled histograms of the absolute error

for all individual samples in the test set. Here, we see that in addition to outperforming the benchmark optics tables and existing parameterization on average, the most extreme errors produced by the ANN are also far smaller than those produced by the existing parameterization. Furthermore, the ANN's histograms tend to have peaks at lower error values than the other methods. Note that because of the log-scaling, the peak represents a large number of samples and the size of the error distribution's tails

is exaggerated. An interesting feature from Figure 4 is that the lookup tables tend to have longer left-tails, representing cases with very low error. These occur because some regions in the input space have little to no variability in the output space, the large regions where extinction is near zero for instance. The linear interpolation in the lookup tables can perfectly fit constant valued functions but the ANN and Chebyshev methods will still have a small amount of error. Ultimately, the key observation from Figure 4 is that the ANN's errors do not have a large right-tail, meaning that even for the input queries where the ANN

performs worst, we still expect very accurate estimates of aerosol optical properties.

Finally, Figure 5 shows a joint-histogram of bulk aerosol optical properties estimated by the existing parameterization and by direct computation with Mie code for all samples in the test set. Separate joint histograms are not included for the ANN outputs, instead a red contour in each of the joint histograms denotes the boundary containing all samples. Notable patterns appear in the joint histograms of the shortwave extinction field and the fields derived from it (SW scattering and SSA), and

405 to a lesser degree the other predicted fields. These arise in the Ghan and Zaveri (2007) parameterization from the Chebyshev polynomial fit used to approximate optical properties as a function of surface mode radius. The Chebyshev polynomials are smooth functions that do not perfectly fit the bulk extinction efficiency curve for instance, and consistently over- or under-shoot it for certain $r_s$ values. Because bulk extinction efficiency is very sensitive to the particle size distribution this effect is obvious in Figure 5.

Drawing the training set from a regular grid over the input space has ensured good coverage of possible input values, while generating a test set of equal size consisting of intermediate values that are not near points in the training or validation data helps

**Table 1.** Mean absolute error for bulk optical property estimates using different methods. Note that only bulk absorption efficiency is computed for the longwave bands and that shortwave single scattering albedo (SSA) and bulk scattering efficiency are computed from shortwave absorption and extinction efficiencies. The overbars denote that these are bulk values integrated over log-normal size distributions (Eq. 1)

| Method | N-Params. | $\overline{Q}_{\text{Abs.}}$ (SW) | $\overline{Q}_{\text{Ext.}}$ (SW) | $\overline{g}$ (SW) | $\overline{Q}_{\text{Sca.}}$ (SW) | $\overline{SSA}$ (SW) | $\overline{Q}_{\text{Abs.}}$ (LW) |
|---|---|---|---|---|---|---|---|
| Random ANN: | $10^4$ | $8.6 \times 10^{-5}$ | $3.6 \times 10^{-4}$ | $1.1 \times 10^{-4}$ | $3.5 \times 10^{-4}$ | $3.2 \times 10^{-4}$ | $3.7 \times 10^{-5}$ |
| Serial ANN: | $10^4$ | $1.1 \times 10^{-4}$ | $4.2 \times 10^{-4}$ | $1.2 \times 10^{-4}$ | $4.1 \times 10^{-4}$ | $4.3 \times 10^{-4}$ | $7.3 \times 10^{-5}$ |
| Ghan and Zaveri (2007): | $10^5$ | $1.8 \times 10^{-2}$ | $2.0 \times 10^{-1}$ | $2.5 \times 10^{-2}$ | $2.0 \times 10^{-1}$ | $5.2 \times 10^{-2}$ | $1.4 \times 10^{-2}$ |
| Lookup Table: | $10^6$ | $3.8 \times 10^{-3}$ | $6.6 \times 10^{-3}$ | $1.7 \times 10^{-3}$ | $9.0 \times 10^{-3}$ | $2.6 \times 10^{-3}$ | $2.5 \times 10^{-3}$ |
| Lookup Table: | $10^7$ | $1.0 \times 10^{-3}$ | $1.9 \times 10^{-3}$ | $5.3 \times 10^{-4}$ | $2.5 \times 10^{-3}$ | $6.8 \times 10^{-4}$ | $6.7 \times 10^{-4}$ |
| Lookup Table: | $10^8$ | $3.1 \times 10^{-4}$ | $7.2 \times 10^{-4}$ | $2.1 \times 10^{-4}$ | $8.6 \times 10^{-4}$ | $2.0 \times 10^{-4}$ | $2.0 \times 10^{-4}$ |
| Lookup Table: | $10^9$ | $1.2 \times 10^{-4}$ | $3.9 \times 10^{-4}$ | $1.1 \times 10^{-4}$ | $4.2 \times 10^{-4}$ | $7.6 \times 10^{-5}$ | $7.6 \times 10^{-5}$ |

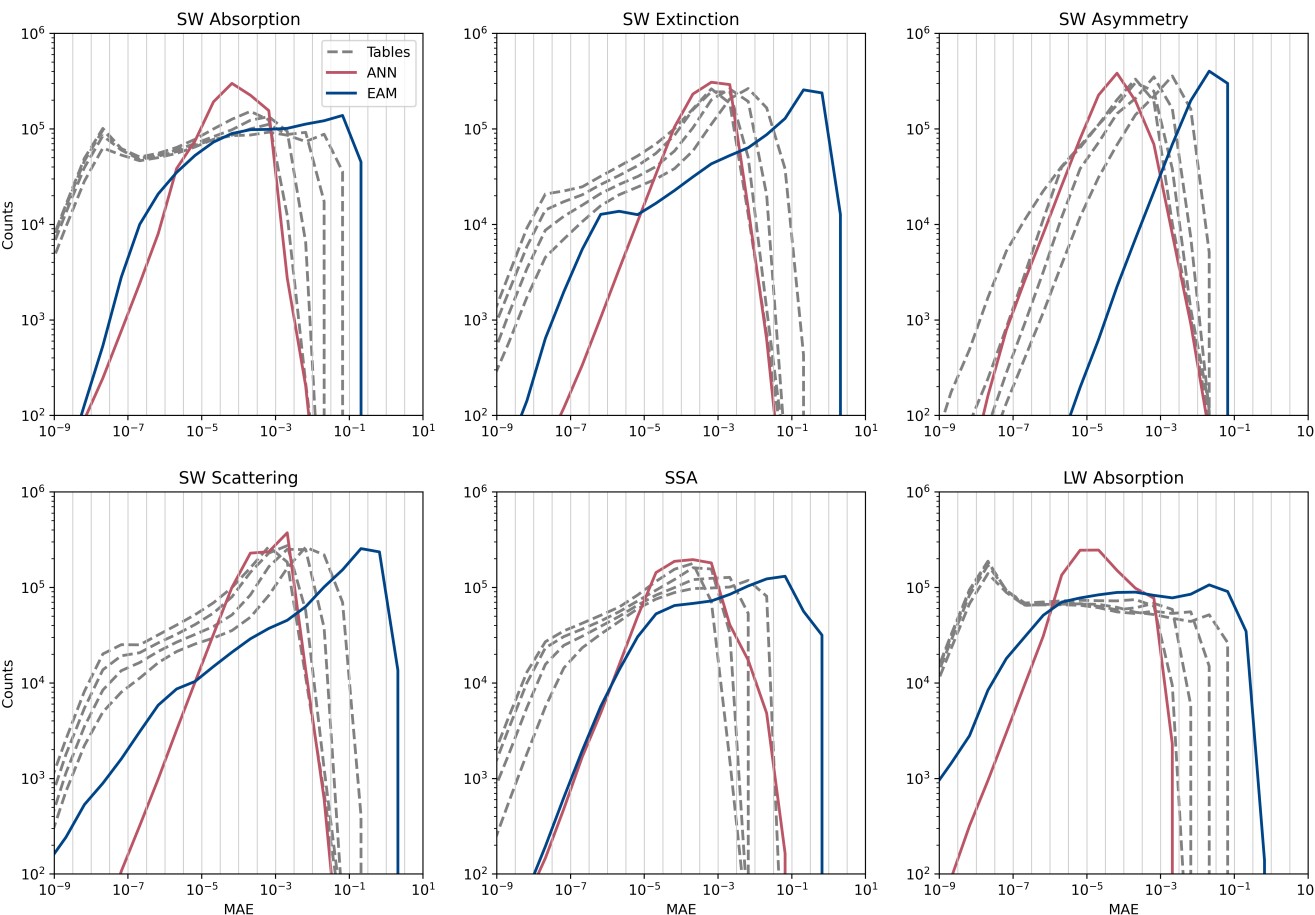

**Figure 4.** Error histograms for estimates of the bulk aerosol optics test dataset. These panels show the distribution of errors on a log-log histogram to make outlier cases with high error more apparent. The vertical grid shows the bin edges of the histogram. The blue and magenta lines represent the Chebyshev polynomial based parameterization and the neural network respectively. The dashed gray lines represent the error from applying linear interpolation to pre-computed optics datasets of varying resolution, with the highest resolution tables appearing to the left and progressively coarser tables to the right.

demonstrate that the ANN will not perform unexpectedly when interpolating within the region defined by the training data. Together, Table 1 and Figures 4 and 5 demonstrate that the ANN parameterization not only provides a dramatic performance improvement over the current approach, but can also be expected to perform exceedingly well for the full range of possible input data, with no extreme cases of high error. The ANN is therefore an accurate and reliable replacement for the current bulk aerosol optics parameterization.

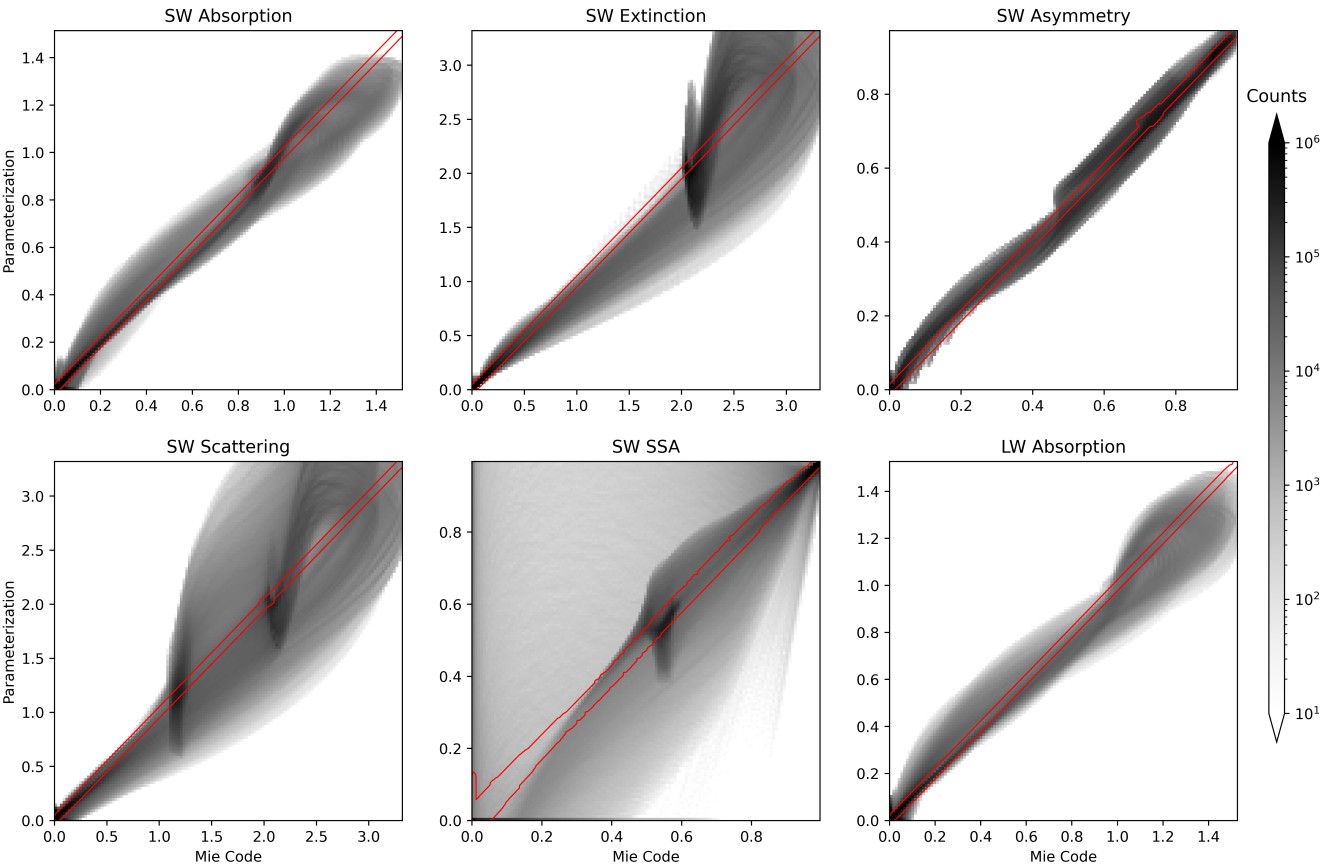

**Figure 5.** Scatter plot-like joint histograms comparing optical properties from the Chebyshev interpolation based parameterization and Mie code. Grey shading indicates the density of data-points. The red contour contains all outputs from the neural network, which all lie very close to the 1-to-1 line.

## 6 Conclusions

This work has demonstrated the effectiveness of machine learning for emulating the aerosol optical properties that are crucial to climate simulation. A neural network is capable of producing bulk optical property estimates that are substantially more accurate than those produced by the existing (Ghan and Zaveri, 2007) parameterization in E3SM and CESM and does so with an order of magnitude smaller memory requirements. The compute requirements for evaluating an ANN with $10^4$ parameters is larger than the compute used by the current approach, but this parameterization is evaluated every time EAM calls radiation code, and evaluating the ANN requires negligible compute compared to the radiation code, so impact on model run-time should be negligible. Additionally, the ANN outperforms lookup-table based optics emulators that resolve aerosol optical properties at much higher resolution than the existing scheme. Testing over a wide range of possible input data showed that the neural network performs well over the possible input space and will not produce any outlier errors or unexpected results within this

range. Representation of aerosol direct effects is a major source of uncertainty in climate simulation, and while representation of aerosol optics is likely only a small component of this uncertainty, adequate representation of this physics is a key step forward towards accurately representing aerosols in general.

This work, to some degree, should be seen as a first step or proof of concept, and a demonstration of the power of randomly wired networks for this problem. Our ultimate goal is to develop a neural network based parameterization that represents core-shell scattering; a physical model that is too computationally expensive to represent with existing parameterizations. While this work presents the machine learning technique and evaluates it directly against Mie code, we expect to follow it with a climate modeling study evaluating the impacts of this parameterization, and a future core-shell scattering model, on E3SM simulations.

In addition to developing a new parameterization, we applied a recently developed (Xie et al., 2019) neural architecture search strategy that randomizes wiring patterns in deep neural networks. Key findings were that deeper ANNs significantly outperformed a single layer ANN of comparable size. Also, the majority of randomly constructed ANN architectures (which include skip connections) outperformed conventional multi-layer perceptron networks. In the context of this study, the NAS allowed us to identify neural architectures that provide a substantial performance improvement with no increase in network

size.

    Our findings provide some insights into ANN design. The fact that the majority of randomly wired networks outperform multi-layer networks with serially connected layers indicates that inclusion of skip connections may be critical for this type of problem. In image processing, convolutional neural networks with a large number of layers and skip connections (He et al., 2016; Huang et al., 2017) were identified as superior to serially connected designs several years ago, and have dominated deep

learning research since. While using skip connections in networks constructed of fully connected layers is certainly not a new idea (Lang and Witbrock, 1988), it has received comparatively little attention in recent machine learning literature. This work indicates that inclusion of skip connections could be an effective way to train smaller regressor and function fitting neural networks to fit complicated data or surfaces.

    To the best of our knowledge this is the first use of randomly wired neural architecture search in the atmospheric sciences.

Their performance against conventional serially connected feed forward ANNs in this task was striking. The majority random wirings were better able to represent Mie optics than serial wirings by a substantial amount (about 10-30% in the shortwave regime and 65% in the longwave) with no increase in model complexity in terms of the number of trainable parameters. There has recently been significant push to leverage new advances in machine learning to replace the various existing parameterizations used by climate and weather models with more performant and/or accurate representations (e.g. Gettelman et al. (2021);

Lagerquist et al. (2021)). Many of these problems, like the Mie optics problem addressed here, are data-rich and well suited for neural architecture search, because training data can be produced by an accurate but computationally expensive numerical simulation. Our results indicate that when using neural networks for this type of application, significant performance improvements can be achieved by taking care to design or select network architectures optimized for the target task. NAS algorithms and random wirings have, so far, received little attention in the Earth sciences, and random network wiring may be a fruitful

strategy for developing neural network based parameterizations and physics emulators in the future.

*Code and data availability.* The code created as part of this research is available from the project's Github repository: `https://github.com/avgeiss/aerosol_optics_ml`, which has been archived with a DOI using Zenodo: `https://doi.org/10.5281/zenodo.6767169`.

Wiscombe's "mivev0" is thoroughly documented in Wiscombe (1979, 1980) and has been preserved in several locations online, including as part of the CESM 1.0 code accessible here: https://www.cesm.ucar.edu/models/cesm1.0/cesm/cesmBbrowser/html_code/cam/miesubs.F.html#MIEV0 (Accessed April $25^{th}$ 2022).

PyMieScatt is available from: `https://github.com/bsumlin/PyMieScatt`. With documentation here: `https://pymiescatt.readthedocs.io/en/latest/`, and can be installed via the pip Python package manager (Accessed Feb $8^{th}$ 2022).

All data produced as part of this study including, optics tables, random ANN files, and Chebyshev coefficients generated by our python port of the Ghan and Zaveri (2007) parameterization has been made available online: `https://doi.org/10.5281/zenodo.6762700`. We note that all the data stored here can be produced by running the code in the project's Github repository.

*Author contributions.* AG wrote manuscript, performed experiments, developed methods and code, PLM: secured funding, conceived project, edited manuscript, provided assistance with Fortran code, JCH: provided input on project planning/direction, BS: provided assistance with Fortran code.

*Competing interests.* Po-Lun Ma is a Topical Editor of Geoscientific Model Development. Other authors declare that they have no conflict of interest.

*Acknowledgements.* This study was supported as part of the "Enabling Aerosol–cloud interactions at GLobal convection-permitting scalES (EAGLES)" project (project no. 74358), sponsored by the U.S. Department of Energy, Office of Science, Office of Biological and Environmental Research, Earth System Model Development (ESMD) program area. The Pacific Northwest National Laboratory is operated for DOE by Battelle Memorial Institute under contract DE-AC05-76RL01830. The research used high-performance computing resources from the PNNL Research Computing and resources of the National Energy Research Scientific Computing Center (NERSC), a U.S. Department of Energy Office of Science User Facility located at Lawrence Berkeley National Laboratory, operated under Contract No. DE-AC02-05CH11231 using NERSC awards ALCC-ERCAP0016315, BER-ERCAP0015329, BER-ERCAP0018473, and BER-ERCAP0020990. We would also like to thank Sam J. Silva and William Yik for their helpful discussions regarding randomly wired neural networks.

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

**Table A1.** Errors between optical properties computed with PyMieScatt and miev0.

| | $Q_{\text{Abs.}}$ | $Q_{\text{Sca.}}$ | $g$ |
|---|---|---|---|
| Max Abs. Err. | $1.8 \times 10^{-3}$ | $1.6 \times 10^{-2}$ | $9.6 \times 10^{-2}$ |
| 99.9 %-ile | $1.1 \times 10^{-3}$ | $1.5 \times 10^{-3}$ | $8.0 \times 10^{-4}$ |
| 99 %-ile | $3.3 \times 10^{-4}$ | $4.3 \times 10^{-4}$ | $4.4 \times 10^{-4}$ |

**Table A2.** Constants used to standardize ANN inputs. For all variables but real refractive index standardization is done after taking the natural logarithm. $1 \times 10^{-6}$ is added to the imaginary refractive index before taking the logarithm.

| $(\mu/\sigma)$ | Real Ref. Ind. | Imaginary Ref. Ind. | Surf. Mode Rad. ($R_{surf}$) | Wavelength ($\lambda$) | $R_{surf}/\lambda$ |
|---|---|---|---|---|---|
| SW | 1.6 / 0.2 | -7.0 / 4.0 | -14.5 / 2.3 | -13.6 / 1.0 | -0.9 / 3.9 |
| LW | 1.7 / 0.3 | -7.0 / 3.9 | -14.5 / 2.3 | -11.5 / 1.1 | -3.0 / 2.5 |