# Peer review of "Emulating Aerosol Optics with Randomly Generated Neural Networks"

_EGUsphere, 2022_

## Author Response (AR1)

Firstly, we would like to sincerely thank the editor and both peer reviewers for their helpful comments. We recognize that peer review is a time-consuming and often thankless job, but your input is essential and has helped improve our updated version of the manuscript. We have provided some comments to both reviewers below, responded point by point to each of the reviewer comments individually, and have submitted an updated version of the manuscript along with tracked changes.

**To both reviewers:**

During the process of responding to the reviewer comments we identified two errors in our code that required correction and ultimately required re-running almost all the experiments in the paper:

- 1) The lookup tables and optics dataset are large, and early in the project these data were stored using 16-bit floating point precision to save space and reduce data loading times. This is not an issue for the analysis of the Ghan and Zaveri 2007 parameterization because its errors are of the order 0.1 for values of the order 1.0 and any floating-point truncation errors are negligible. The accuracy of the best neural network models and largest optics lookup table evaluated in the study approach the limits of 16-bit precision however. Remedying this required re-running the experiments for the full study using data stored at higher, 32-bit, precision. The main impacts of re-running the experiments were slightly lower error for the neural network models across the board. The training procedure had to be slightly changed to include an extra round of training after an additional learning rate reduction. The random network generation was re-run so different optimal random architectures were found and Figure 3 had to be updated to reflect this.
- 2) A scaling factor of either 1.4 or 1.1 (depending on aerosol mode) was not correctly applied when computing the physical values analyzed in Figures 4 and 5 and Table 1. The factor arises from the coefficient in front of the integral in Equation 1 in the revised manuscript. It is omitted in the existing parameterization code because it is later canceled by another operation, but should have been included for reporting the error metrics in our manuscript. This error caused an artifact to appear in Figure 5 in the initial draft and caused the MAEs reported for all methods in Table 1 to be slightly overstated. Though they were increased by the same factor for all methods, so our comparison and discussion of the methods' relative performance in the text remains mostly unchanged.

After fixing these issues, the paper's conclusions and main results remained unchanged, but the content of most tables and figures are slightly altered. Because the entire project had to be re-run, we took the opportunity to make some changes that help address some of the reviewers' concerns and some unsolicited changes that, we believe, increased the quality of the study:

- 3) We have generated a much larger testing dataset. It has about 1000x the number of samples used in the initial submission and is of comparable size to the combined training and validation datasets. Furthermore, the testing set is gridded to provide full coverage of the potential input space and does not include any points potentially very close to training samples (the previous testing set was randomly generated).
- 4) We omitted the second round of training for the selected random architectures that included the samples from the validation set. We found that this step did not significantly alter their performance.

- 5) The training procedure description in Section 4.4 was re-written in terms of epochs instead of total number of batches and modified so that the ANNs could fit the higher-accuracy training data by including an additional epoch at a lower learning rate at the end of training.
- 6) The design of Figure 4 was changed. The histogram bins had previously used a linear scaling which assigned most of the small error cases to the lowest bin. Now the bins use log-scaling, and the bin edges are clearly marked in the figure.
- 7) Equation 1 was added to Section 3.5 that shows exactly how the bulk aerosol optical properties can be calculated using Mie code. We have made the language referencing bulk optical properties and their descriptions clearer.

**Responses to reviewer 1 comments:**

This manuscript describes the development of neural networks to replace the aerosol optics in a climate model with a more detailed treatment, which is based on running the same types of codes with more detail to develop a parameterization which better represents more detailed modeling. In general the manuscript is well written and clear. There is a nice discussion of how the neural network is developed. However, my main critique is that the evaluation section (section 5) is pretty minimal. Just some error curves. What does it look like in the full model? You have demonstrated that the new parameterization represents the more detailed code better than the existing parameterization. Does it change the answers in the climate model it is designed for in any meaningful way, and does it cost anything more to run it. Also good to note in the conclusions what lessons you learn from this experience about building neural networks for parameterization replacement. This is probably suitable for publication with minor revisions, but with at least trying it in a climate model perhaps.

We have expanded Section 5 and added more discussion and evaluation of the ANN against additional optical parameters (scattering and single scattering albedo). We also added some discussion of lessons learned and of the implications of the random wirings and the potential importance of skip connections in applications like this one to Sections 4.5 and 6. This is an ongoing project however, and our ultimate goal is to use the random ANNs and the approach to training used here to develop parameterizations capable of more complex optics than what is currently available for climate modeling (core-shell models). Our plan is to publish a follow-on paper that evaluates climate modeling results using multiple ML-based parameterizations. Because the random ANN approach is fairly involved and a significant result on its own with implications for future development of ML-based parameterizations, we thought it was best to publish this manuscript separately describing the random ANNs in detail and evaluating them directly against Mie scattering models.

**Page 1, L10: Would be good to have more detail on what 'outperform' means specifically in another sentence or two.**

Added this on line 10: "Finally, the ANN-based parameterization produces significantly more accurate bulk aerosol optical properties than the current parameterization when compared to direct Mie calculations using mean absolute error"

Page 1, L15: Disingenuous. The direct effects of aerosols are not the largest uncertainty: only indirect effects on clouds.

We have updated this to be clearer (line 18): "They have long been known as one of largest sources of internal uncertainty in climate modeling, primarily due to cloud interactions, but with a significant contribution from direct effects as well (Bellouin et al., 2020)."

*Page 2, L29: Example of climate models generating training data (for replacing part of a parameterization: Gettelman et al 2021.*

Gettelman, A., D. J. Gagne, C.-C. Chen, M. W. Christensen, Z. J. Lebo, H. Morrison, and G. Gantos. "Machine Learning the Warm Rain Process." Journal of Advances in Modeling Earth Systems 13, no. 2 (2021): e2020MS002268. https://doi.org/10.1029/2020MS002268. Thanks, noted. (line 32)

Page 3, L63: clarify 'these optical properties (absorption...etc'

Updated line 67: "(bulk absorption, extinction, and asymmetry parameter)"

Page 3, L77: what is the size range here? Please be explicit.

We have added clarification to line 75: "significant portion of atmospheric aerosols have size parameters  $(x = 2\pi r/\lambda)$  within the Mie regime, particularly in the shortwave radiative bands used by EAM's radiative transfer code. There is no strict definition of the bounds of the Mie regime, but typically one would use Mie code to estimate optical properties for size parameters within about 2 orders of magnitude of unity and geometric or Rayleigh approximations for larger or smaller particles (respectively) depending on the accuracy required for the application. Here we use a Rayleigh approximation for size parameters less than 0.05 and Mie code for everything larger."

Page 3, L90: The CESM reference should probably be Danabasoglu, et al 2020.

Danabasoglu, G., J.-F. Lamarque, J. Bacmeister, D. A. Bailey, A. K. DuVivier, J. Edwards, L. K. Emmons, et al. "The Community Earth System Model Version 2 (CESM2)." Journal of Advances in Modeling Earth Systems 12, no. 2 (2020): e2019MS001916. https://doi.org/10.1029/2019MS001916.

Noted, line 97.

Page 5, L148: how much error is there in the approximations? Can you quantify it?

Yes, it is shown in Table 1. We added a note of this on line 153.

Page 9, L251: why would a random network do better? Is there an explanation? Isn't that a form of overfitting?

We added discussion in Section 4.5 and 6. In short, the fact that most random nets outperform their conventional counterparts indicates that skip connections are important for this problem. There is fairly detailed research on why skip connections are useful for deep learning in the context of convolutional neural networks, but they are only rarely used (at least for now) in ANNs with fully connected layers. Finally, some of the random nets are bound to be worse than others and the random search helps us find a good architecture for the problem and balance the model size vs performance.

Overfit is a significant concern with this level of model optimization, so we have taken great care to generate a validation and testing procedure that would make a potential overfit obvious. In our revised manuscript we have added evaluation of the best conventional ANNs on the test set as a row to table 1 and reported the values of the best random ANNs evaluated on the test data using the same method as in Figure 2 to section 4.4. Both these checks help make clear that validation set performance is predictive of test set performance.

**Page 10, L289: Please describe these terms a bit. ReLU, ELU, Leaky ReLU and Parametric ReLU**

We have updated this to use more descriptive names and provided relevant citations (line 311).

Page 10, L291: what is a transfer function? Above you call them activation functions. Please clarify.

We have updated the manuscript to use consistent terminology throughout (we were using these interchangeably)

Page 11, Figure 2: is the red dot the 'optimum' network?

Yes. We have added a note of this to the caption.

Page 12, L350: can you make the evaluation a bit more quantitative in spots? It seems a bit 'weak' right now, especially compared to the rest of the paper.

We have added evaluation of extinction efficiency and single scattering albedo, and some additional discussion to Sections 5 and 6.

**Page 13, L360: What does the first column (Table:) of table 1 mean? Should it say something?**

These represent using a lookup table with linear interpolation instead of the neural network. We have added some clarification to the manuscript.

**Page 13, L364: Can you explain the patters in Figure 5? What do they arise from?**

These arise from the Chebyshev polynomial fit that the parameterization uses to approximate optical properties as a function of mode radius. The method only uses a 5th order Chebyshev fit to represent the variability along this dimension, and in the cases that the fit is not very good, the Chebyshev polynomial will over- and under-shoot the true curve. Because the scattering efficiency is very sensitive to the mode radius this over/undershoot is very obvious in Figure 5. Discussion was added at line 405

**Page 16, L402: Other lessons learned? It would be great to share in the paper.**

More discussion was added to Sections 4.5 and 6. Also note that there are some comments at the end of Section 4.3 on transfer functions and regularization techniques that we decided not to use early on in the project because they were clearly performing worse.

**Responses to reviewer 2 comments:**

The manuscript (MS) presents a new method (based on Artificial Neural Networks (ANNs)) for online calculation of the optical properties of the internally mixed aerosols. Current parametrizations and lookup tables are either computationally unaffordable or fail to capture the large variabilities in aerosol properties. The training dataset is based on the Mie code that directly computes the optical properties of aerosols by considering the variability of the particle sizes, wavelengths, and refractive indices. This approach is similar to previous parameterizations but uses a higher resolution for different parameters. By evaluating ANNs with randomly generated wirings, the optimal network architectures are identified for SW and LW. The results show that randomly generated deep ANNs lead to lower error compared to the conventional multi-layer perceptron. Besides, the ANN-based parameterization outperforms the current parameterization.

The paper is very well structured and written. I really enjoyed the detailed explanations of the assumptions and methods that makes it easy to follow the results. The methods and results are robust with major benefits for the aerosol modeling community. Thus, I recommend publication after addressing the minor points/questions listed below.

With respect to the I/O, it is not clear why nine variables are chosen. Any pre-processing or input selection procedure? Especially two parameters "surface mode radius over wavelength" and "surface mode radius" are obviously correlated. This should not happen.

These I/O variables are all used by the existing parameterization except the surface mode radius over wavelength. It often helps ANNs to hand-construct input features that are non-linear combinations of other inputs. In this case we know that size parameter is very relevant to Mie scattering, so this is the one hand-made feature we opted to add. In our early experiments for this project, it helped with faster training and more accurate models. We added a note about this on line 220.

**Why do you need one-hot encoding? What additional information does it contain for the modes?**

One-hot encoding is a common way to encode categorical data for neural networks. It is usually easier for them to learn with these one-hot vectors than using a scalar input that can have 4 different values (also consider that a scalar input implies that some categories are closer than others, e.g. modes 2 and 3 are closer than 1 and 4, and that may not be true). Each mode assumes a different log-standard-deviation so that is the main information that is encoded here. There may be other information about the behavior of each mode that the ANN learns to infer from the one hot encoding too. We have added more discussion of the input parameter choices starting on line 217.

**It can be expected that 2-3 hidden layers can capture the nonlinearities of the system very well and more hidden layers often lead to over-fitting (shown in Fig 2). But it is not clear if the rather minor MAE reduction by random wired networks (outperform is too strong here) is justified by its computational costs/complexity.**

Please note that the horizontal axis in Figure 2 represents total parameter counts for the model. The random networks are compared to serially connected ANNs with similar parameter counts, so while there is added complexity in terms of the network graph the number of floating-point operations to evaluate the random nets is not higher. While the improvement of random ANNs over conventional ANNs is small compared to the difference between the current parameterization and any ANN, the random ANNs provide an additional 10-30% reduction in error in the shortwave regime and 65% in the longwave regime

at no additional computational cost during use. Also note that the lines in Figure 2 represent the best performing conventional ANNs after 5 trials of re-training from scratch, and the majority of the dots, which represent each individual random network, have lower MAE than the conventional ANNs of the same size. Not only do the best random nets outperform the conventional ANNs but the majority do. We have added more discussion on this topic to Sections 4.5 and 6.

**I would like to see ANN vs. Mie similar to figure 5 but for all parameters: extinction coefficient, single scattering albedo and asymmetry parameter in SW and WL.**

We have added panels to Figure 5 that include bulk SW scattering efficiency and single scattering albedo. The parameterization only handles absorption in the longwave.